# P-21 Kinase 1 or 4 Knockout Stimulated Anti-Tumour Immunity Against Pancreatic Cancer by Enhancing Vascular Normalisation

**DOI:** 10.3390/ijms26178357

**Published:** 2025-08-28

**Authors:** Arian Ansardamavandi, Chelsea Dumesny, Yi Ma, Li Dong, Sarah Ellis, Ching-Seng Ang, Mehrdad Nikfarjam, Hong He

**Affiliations:** 1Department of Surgery, Austin Precinct, The University of Melbourne, 145 Studley Rd, Heidelberg, VIC 3084, Australia; arian.ansardamavandi@student.unimelb.edu.au (A.A.); watsoncj@unimelb.edu.au (C.D.);; 2Department of General Surgery, Monash Health, Clayton, VIC 3806, Australia; 3Department of Anatomy & Physiology, University of Melbourne, Melbourne, VIC 3010, Australia; 4School of Cancer Medicine, La Trobe University, Melbourne, VIC 3000, Australia; 5Olivia Newton-John Cancer Research Institute, Heidelberg, Melbourne, VIC 3086, Australia; 6Mass Spectrometry and Proteomics Facility, Bio21 Molecular Science and Biotechnology Institute, University of Melbourne, Parkville, VIC 3010, Australia; 7Department of Hepatopancreatic-Biliary Surgery, Austin Health, 145 Studley Rd, Heidelberg, VIC 3084, Australia

**Keywords:** PAK1, PAK4, pancreatic ductal adenocarcinoma (PDA), vascular normalisation, immune activation

## Abstract

Pancreatic ductal adenocarcinoma (PDA) exhibits diverse molecular aberrancies that contribute to its aggressive behaviour and poor patient survival. P-21-activated kinase 1 (PAK1) and PAK4 drive the tumorigenesis of PDA. However, their roles in tumour vasculature and the impact on immune response are unclear. This study aims to investigate the effects of PAK1 and PAK4 on tumour vasculature, immune cell infiltration, and the connection between using PAK1-knockout (KO), PAK4 KO, and wild-type (WT) PDA cells in cell-based and mouse experiments. Tumour tissues isolated from a syngeneic mouse model were immuno-stained to determine the changes in tumour vasculature and immune cell infiltration/activation, followed by a proteomic study to assess biological processes involved. PAK1KO or PAK4KO suppressed tumour growth by reducing angiogenesis while enhancing vascular normalisation, enhanced the infiltration/activation of T-cells and dendritic cells associated with upregulation of ICAM-1 and VCAM-1 in the tumour microenvironment, and stimulated vascular immune crosstalk via an ICAM-1-mediated mechanism. This was supported by proteomic profiles indicating the regulation of endothelial cell and leukocyte trans-endothelial migration in PAK1- or PAK4-knockout tumours. In conclusion, PAK1KO or PAK4KO enhanced tumour vascular normalisation while reducing angiogenesis, stimulating immune cell infiltration and activation to suppress tumour growth.

## 1. Introduction

Pancreatic ductal adenocarcinoma (PDA) is the sixth leading cause of cancer-related deaths worldwide, and its mortality is expected to increase to 300,000 in 2030 [1,2]. It has a 5-year survival rate of 11% despite advancements in surgical techniques and chemotherapeutic treatments [3]. The low survival rate of PDA originates from its late diagnosis, intrinsic therapy resistance, and an aggressive tumour microenvironment (TME) [4]. Current treatments, such as FOLFIRINOX and gemcitabine-based combinations, offer only modest survival benefits [5]. Immunotherapy has largely failed to demonstrate clinical efficacy in PDA due to the immunosuppressive TME [6].

Within the TME of PDA, characterised by a dense stroma and immune suppression, tumour blood vessels are structurally abnormal and functionally impaired, often displaying disorganised architecture, excessive permeability, and insufficient pericyte coverage. These vascular abnormalities not only hinder effective drug delivery but also contribute to immune evasion, resistance to therapies and inhibition of anti-tumour immunity [7,8]. While tumour vasculature is essential for supporting tumour growth, conventional anti-angiogenic therapies can paradoxically induce drug resistance and enhance tumour invasiveness [9]. As a result, strategies aimed at normalising the tumour vasculature by modulating the TME have emerged as a promising approach to improve both anti-angiogenic and anti-tumour responses [10,11].

P21-activated kinases (PAKs) have emerged as promising therapeutic targets in PDA. PAKs are a family of serine/threonine kinases that act downstream of KRAS, a gene mutated in over 90% of PDA cases. The PAK family comprises six isoforms, classified into two groups based on sequence homology and regulatory mechanisms: group I (PAK1-3) and group II (PAK4-6) [12,13]. Among the PAK isoforms, PAK1 and PAK4 are the most extensively studied in cancers [14]. In PDA, PAK1 has been shown to inhibit cancer cell apoptosis and suppress intra-tumoral infiltration of CD4^+^ and CD8^+^ T-cells [15]. PAK4 plays a multifaceted role in cancer progression by regulating cytoskeletal dynamics, promoting cancer cell survival, enhancing stemness, and modulating the tumour immune microenvironment through suppression of T-cell-mediated responses [16]. More recently, PAK4 has been shown to suppress T-cell responses in melanoma, prostate cancer, and glioblastoma. These findings align with increased intra-tumoral CD8^+^ T-cell infiltration observed in a PDA mouse model treated with the PAK4 inhibitor PF-3758309 [17,18,19,20,21]. While the immunomodulatory roles of PAK1 and PAK4 in PDA have been investigated, the impact of PAK on tumour vasculature and how these vascular changes influence the immune system, including T-cells and dendritic cells, remains poorly understood. We have explored the effects of PAK1 and PAK4 on the tumour vasculature of PDA and its impact on the PAK-regulated tumour immune response in this study.

## 2. Results

### 2.1. PAK-Knockout Suppressed Pancreatic Tumour Growth by Reducing Angiogenesis and Promoting Tumour Vasculature Normalisation

Knockout of PAK1 or PAK4 suppressed pancreatic tumour growth in a syngeneic mouse model, demonstrated by reduced tumour volume and tumour weight (Figure 1A–C,J–L and Appendix A). PAK1KO, as well as PAK4KO, decreased the number of tumour vessels, as shown by reduced micro-vessel density (MVD) measured by H&E staining of tumour tissues (Figure 1D,M). Knockout of PAK1 or PAK4 inhibited angiogenesis within the tumour by decreasing the intra-tumoral expression of CD31 (Figure 1E,N) and CD34 (Figure 1F,O), two common endothelial cell markers responsible for angiogenesis. Decreased angiogenesis by either PAK1KO or PAK4KO further increased hypoxia in tumours, showing increased HIF-1α staining in the PAK1KO or PAK4KO tumour tissues compared to the WT tumour tissues (Figure 1G,P).

Inhibition of angiogenesis restrains tumour growth but may also cause hypoxia that renders cancer cells resistant to chemo- and radiotherapies and compromises anti-tumour immunity. This paradox can be resolved by vascular normalisation, which is indicated by increased pericyte coverage measured by the increased ratio of NG2 or α-SMA, two markers of mature pericytes, to CD31, a marker of endothelial cells [22]. Knockout of PAK1 or PAK4 increased the pericyte coverage in pancreatic tumour tissues by increasing the ratios of NG2 to CD31 (Figure 1H,Q) and of α-SMA to CD31 (Figure 1I,R). These results indicate that deletion of PAK1 or PAK4 inhibited tumour angiogenesis while increasing the normalisation of the tumour vasculature, contributing to reduced pancreatic tumour growth.

### 2.2. PAK Knockout Stimulated T-Cell Infiltration Associated with Enhanced Activation of Dendritic Cells

Normalisation of the tumour vasculature enhances perfusion, improving immune cell infiltration. As described above, knockout of PAK1 or PAK4 enhanced tumour vasculature normalisation by increasing pericyte coverage (Figure 1H,I,Q,R). We further determined the immune cell infiltration. PAK1KO increased the infiltration of CD3^+^, CD4^+^, and CD8^+^ T-cells (Figure 2A–C). PAK1KO also increased intra-tumoral CD103^+^ and CD11b^+^ dendritic cells (Figure 2D,F) but decreased CD11c^+^ dendritic cells (Figure 2E). PAK1KO further promoted the activation of dendritic cells by increasing the expression of CD86^+^, CD40^+^, and MHC-I^+^ cells (Figure 2G–I). However, PAK1KO reduced the expression of MHC-II^+^ cells (Figure 2J), indicating that PAK1 differentially affected the types of dendritic cells. PAK4KO also stimulated the infiltration of CD3^+^, CD4^+^, and CD8^+^ T-cells (Figure 3A–C) while increasing intra-tumoral CD103^+^, CD11c^+^ and CD11b^+^ dendritic cells (Figure 3D–F). Moreover, PAK4KO enhanced the activation of dendritic cells by promoting the expression of CD86^+^, CD40^+^, MHC-I^+^, and MHC-II^+^ (Figure 3G–J). Interestingly, the levels of MHC-I and MHC-II in PAK1KO and PAK4KO tumour cells were unchanged (Figure 3K). These results suggest that knockout of PAK1 or PAK4 enhanced dendritic cell maturation and activation, contributing to intra-tumoral T-cell infiltration.

### 2.3. PAK Knockout Stimulated Cytotoxic T-Cell Infiltration and Activation by Upregulation of ICAM-1 and VCAM-1

Increased expression of endothelial adhesion molecules, intercellular adhesion molecule 1 (ICAM-1), and vascular cell adhesion molecule 1 (VCAM-1) stimulates intra-tumoral lymphocytes infiltration and thus promotes anti-tumour immunity [23,24]. PAK1KO increased the expression of ICAM-1 by 7 times and of VCAM-1 by 10 times in tumour tissues, showing increased density of ICAM-1 and VCAM-1 staining (Figure 4A,B). Tumour cells also express ICAM-1 or VCAM-1, which may not stimulate anti-tumour immunity [25]. PAK1KO did not affect the expression of ICAM-1 by pancreatic cancer cells but increased the expression of VCAM-1 by pancreatic cancer cells about two times (Figure 4C), suggesting that PAK1KO stimulated the expression of ICAM-1 of non-tumour stromal cells rather than the tumour cells, and that PAK1KO increased the expression of VCAM-1 of non-tumour stromal cells more than that of tumour cells. PAK4KO enhanced the expression of ICAM-1 and VCAM-1 in tumour tissues (Figure 5A,B) by 3 and 4 times, respectively, but not by the pancreatic cancer cells (Figure 4C), indicating that PAK4KO specifically stimulated the expression of ICAM-1 and VCAM-1by the non-cancer stromal cells.

The expression of ICAM-1 and VCAM-1 by both endothelial cells and T-cells plays a key role in immune response. ICAM-1 is particularly crucial for T-cell-mediated immune response. We determined the expression of ICAM-1 by endothelial cells (CD31); cytotoxic T-cells (CD8) and dendritic cells (CD103) by multiplex immunostaining. PAK1KO increased the numbers of ICAM-1^+^CD31^+^ (endothelial cell); ICAM-1^+^CD8^+^ (cytotoxic T-cell); and ICAM-1^+^CD103^+^ (dendritic cell) cells (Figure 4D–F). PAK4KO increased the numbers of ICAM-1^+^CD8^+^ and ICAM-1^+^CD103^+^ cells (Figure 5D,E) and showed an increase in ICAM-1^+^CD31^+^ cells; though not reaching statistical significance (Figure 5C). Furthermore, PAK4KO increased the number of CD8^+^CD103^+^ cells; a type of specific active cytotoxic T-cells (Figure 5F); while PAK1KO did not significantly increase the number of CD8^+^CD103^+^ cells (Figure 4G). These results indicated that knockout of PAK1 or PAK4 stimulated cytotoxic T-cell infiltration/activation by upregulation of ICAM-1 and VCAM-1 (Appendix A).

### 2.4. PAK Knockout Promoted Leukocyte Trans-Endothelial Migration Associated with Stimulating Endothelial and Dendritic Cells

The above results indicated that knockout of PAK1 or PAK4 enhanced tumour vascular normalisation and stimulated intra-tumoral T-cell infiltration and activation by upregulation of ICAM-1 and VCAM-1. To determine molecular changes involved, we conducted a proteomic study to compare the protein profiles of PAK1KO tumours, PAK4KO tumours, with WT tumours. Consistent with the results obtained from immunostaining of tumour tissues, from the analysis of the global protein profile, the expression of ICAM-1, VCAM-1, CD11b (a subset of dendritic cells), and MHC class I molecules (H2-D1 and H2-K1) were increased in PAK1KO tumour (Figure 6C) while CD31 and CD34, CD11c (a subset of dendritic cells), and MHC class II molecules (H2-Ab1, H2-Eb1, H2-DMa, and H2-DMb1) were significantly downregulated in PAK1KO tumours (Figure 6C). Protein–protein interaction network analysis of global proteomic data suggested significant alterations in processes involved in endothelial cell migration, dendritic cell function, and MHC-II antigen presentation, and enrichment of the KEGG leukocyte trans-endothelial migration pathway (Figure 6B). The differentially expressed proteins were demonstrated in a volcano plot (Figure 6A).

Similarly, the analysis of the global protein profile indicated that the expression of ICAM-1 was increased in PAK4KO tumours (Figure 7C). PAK4KO also upregulated the expression of CD11c and CD11b (two subsets of dendritic cells) and MHC class I and II molecules (Figure 7C). Protein–protein interaction network analysis suggested upregulated molecules involved in antigen processing and presentation, dendritic cell and leukocyte trans-endothelial migration (Figure 7B), and alterations in endothelial cells. These data confirmed our findings that knockout of PAK1 or PAK4 upregulated ICAM-1-mediated pathway, associated with upregulation of dendritic cells and leukocyte trans-endothelial migration, which may contribute to enhancing vascular normalisation, T-cell infiltration, and activation.

## 3. Discussion

The individual roles of PAK1 and PAK4 in tumorigenesis and immune escape in PDA are recognised in the literature [14,26]. However, the mechanisms by which PAK1 and PAK4 influence immune activation and reshape the TME remain poorly understood [27]. The established roles of PAK1 and PAK4 in regulating intra-tumoral T-cell responses suggest that these kinases may also influence the tumour vasculature [11]. Effector T-cell infiltration is tightly controlled by vascular integrity and endothelial activation, as the tumour vasculature co-evolves with the immunosuppressive microenvironment [18,28]. In this study, we demonstrate that knockout of PAK1 and PAK4 reprograms the immune microenvironment by promoting the activation and infiltration of immune cells, a process associated with normalisation of tumour vasculature.

Knockout of PAK1 and PAK4 suppressed pancreatic tumour growth, which was associated with a reduction in tumour vascularisation (Figure 1A–F,J–O). This reduction in vascularity likely limited nutrient and oxygen supply to the tumour, thereby contributing to tumour regression. The decreased MVD was further evidenced by reduced expression of CD31 and CD34. Decreased vascular density was accompanied by increased tumour hypoxia (Figure 1G,P), indicating impaired perfusion following PAK1 or PAK4 knockout. However, the increased ratios of NG2^+^/CD31^+^ and αSMA^+^/CD31^+^ in PAK1KO and PAK4KO tumours indicated that, despite the overall reduction in vessel density, the remaining vessels exhibited increased pericyte coverage, suggestive of vascular normalisation [22,29] (Figure 1H,I,Q,R). The balance between vascular regression and normalisation dictates the efficacy of treatments and tumour immune response, as abnormal tumour vasculature impedes T-cell infiltration into the tumour [11,30,31,32], impairs drug perfusion and delivery, and facilitates cancer cell escape through leaky vessels [33,34]. The fact that knockout of PAK1 or PAK4 reduced tumour vascularisation, thus vascular regression, while enhancing vasculature normalisation, indicates a key role of PAK1 or PAK4 in balancing vascular regression and normalisation. Previous studies have reported a role of PAK1 in endothelial cell growth and migration [35], which could contribute to a pathological angiogenesis in cancer. In this study, PAK1 was depleted in tumour cells rather than endothelial cells. The reduced vessel density observed in PAK1KO tumours likely reflects impaired paracrine signalling from cancer cells that normally activate PAK1-dependent endothelial pathways.

Vascular endothelial cadherin (VE-cadherin), a predominant component of endothelial adherent junctions, regulates endothelial barrier function and leukocyte transmigration. VE-cadherin plays a key role in vasculogenic mimicry (VM) which is originated from tumour cells, forming a non-classical type of angiogenesis, promoting tumour growth and metastasis. PKA1 activity mediates VE-cadherin phosphorylation regulating intercellular junctional integrity [36]. In pancreatic cancer patients, the expression VE-cadherin is positively corelated with VM, and VM positive is connected to reduced survival of pancreatic cancer [37]. VM, intussusceptive angiogenesis, and vessel co-option contribute to pancreatic cancer progression and therapy resistance [38,39]. VE-cadherin cooperates with EphA2 to activate downstream proteases including MMP14 and MMP2, which are central mediators of VM formation [40]. Our data from the global proteomic study showed that knockout of PAK1 or PAK4 downregulated VE-cadherin (CDH5), EPHA2, CD34, MMP2, and MMP14 proteins that are involved in VM formation (Appendix A), suggesting roles of PAK1 and PAK4 in regulation of VE-cadherin and VM. These findings suggest that PAK inhibition may attenuate not only endothelial angiogenesis but also VM-driven vascularisation, and position PAK signalling as a central regulator of multiple vascularisation programs in pancreatic cancer, underscoring the need for future studies.

We demonstrated that PAK1KO and PAK4KO tumours promoted T-cell infiltration, which was associated with enhanced dendritic cell activation (Figure 2 and Figure 3). In PAK1KO tumours, increased expression of CD103^+^ and CD11b^+^ cells indicated the presence of specific dendritic cell subsets [41,42]. The elevated levels of CD40, CD86, and MHC-I confirmed the increased dendritic cell maturation and activation in PAK1KO tumours. However, a reduction in CD11c^+^ dendritic cells and MHC-II expression was observed in PAK1KO tumours, which was probably due to a compensatory mechanism, where PAK1 deletion led to upregulated PAK4 expression (Figure 4C), decreasing CD11c^+^ dendritic cells and reducing MHC-II expression; PAK4 deletion does not affect PAK1 expression in PAK4KO cell lines. (Figure 4C). The MHC class I and class II antigen presentation pathways are differentially regulated during dendritic cell maturation, reflecting distinct roles in activating cytotoxic and helper T-cell responses, respectively [43]. PAK4KO tumours exhibited more pronounced regression, and not only mirrored the activation of the markers mentioned above but also showed a significant increase in CD11c^+^ and MHC-II^+^ dendritic cell subsets, highlighting a distinct role for PAK4 in suppressing dendritic cell-mediated immune responses, e.g., the requirement for CD4^+^ T-cell maturation and function [44,45]. In addition, activation of CD4^+^ T lymphocytes through immune checkpoint pathways has been shown to promote vessel normalisation, highlighting the synergistic effect between blood vessels and T-cell-mediated immunity in the tumour microenvironment [46,47,48]. Our data from proteomic analysis revealed an increase in Ly6c1 expression in PAK4KO tumours whereas PAK1 deletion did not change the expression of Ly6C (Appendix A). These findings suggest distinct myeloid dynamics across the models: PAK4 deletion preferentially enriched for Ly6C^+^ monocytes. Notably, Ly6C^+^ monocytes are functionally associated with CD11c^+^CD11b^+^Ly6C^+^ populations, which act as key antigen-presenting cells supporting antitumour immune responses [49].

The observed increase in T-cell and dendritic cell activation, along with promoted vascular normalisation in PAK1KO and PAK4KO tumours, was accompanied by elevated expression of ICAM-1 and VCAM-1 (Figure 4 and Figure 5). These endothelial adhesion molecules are critical for mediating leukocyte adhesion, transmigration, and infiltration into the tumours, thereby facilitating effective immune surveillance and anti-tumour immunity [50]. In particular, ICAM-1 plays a key role in the trans-endothelial migration of T-cells [51]. Notably, despite the increased ICAM-1 expression observed in tissue sections, its overall expression levels remained the same across WT, PAK1KO, and PAK4KO cancer cell lines, suggesting that the upregulation primarily originates from the tumour stromal cells (Figure 4C) within the TME. Furthermore, multiplex IHC revealed the distinct co-localisation patterns of ICAM-1 with CD31, CD8, and CD103 in PAK1KO and PAK4KO tumours, indicating enhanced vascular activation and increased activation of immune cells, such as cytotoxic CD8^+^ T-cells and CD103^+^ dendritic cells. Interestingly, PAK4KO also stimulated the intra-tumoral CD103^+^CD8^+^ T-cells, a type of active cytotoxic T-cell that is positively correlated with better prognosis of pancreatic cancer patients [52]. These findings suggest that knockout of PAK1 or PAK4 facilitates vascular–immune crosstalk, stimulating the anti-tumour immunity via an ICAM-1-mediated mechanism.

The data from the analysis of the proteomic profiles of PAK1KO and PAK4KO tumours have revealed that knockout of PAK1 or PAK4 upregulated ICAM-1, MHC class I molecules and enriched the expression of proteins involved in leukocyte trans-endothelial migration. PAK4KO also increased CD11b and CD11c expression associated with the enrichment of dendritic cell pathways (Figure 7). PAK4KO induced the enrichment of KEGG antigen processing and presentation pathways. These results have further confirmed the effects of PAK1 and PAK4 on the tumour immune microenvironment and suggested the regulatory effects of PAK1 or PAK4 on tumour vasculature.

Our findings highlight the distinct roles of PAK1 and PAK4 in modulating the tumour vasculature and immune microenvironment in PDA. PAK1KO or PAK4KO increased the tumour infiltration and activation of immune cells, such as T-cells and dendritic cells, which was associated with enhanced tumour vasculature normalisation. This immune reprogramming and tumour vascular normalisation reciprocally affect each other, reconditioning the tumour immune microenvironment to induce a persistent anti-tumour immunity that regresses tumour growth. Future studies are warranted to validate these effects in orthotopic PDA models and KPC mice.

## 4. Materials and Methods

### 4.1. Cell Lines and Cell Culture

The murine pancreatic cancer cell lines used in this study included wild-type (WT), PAK1-knockout (PAK1KO) and PAK4-knockout (PAK4KO) cells. WT and PAK1KO cell lines were isolated from KPC PAK1^+/+^ and PAK1^−/−^ mice as previously described [15]. KPC PAK4KO cell lines were established from WT KPC cells using the CRISPR-CAS9 knockout technique as described in our previous publication [53]. Cancer cells were maintained in Dulbecco’s Modified Eagle’s Medium (DMEM) with 5% fetal bovine serum (FBS) (Hyclone Laboratories, Melbourne, Australia) and incubated at 37 °C in a humidified chamber containing 5% CO_2_.

### 4.2. Animal Studies

All mouse experiments were conducted with the approval of the Austin Health Animal Ethics Committee (ethics numbers: A2022-05797 and A2023-05849). C57BL/6 mice were housed in the Austin Health Bioresource Facility, where they were routinely monitored for general health parameters. Subcutaneous injections of KPC WT, PAK1KO, or PAK4KO cells (0.5–1 × 10^6^ cells in 100 μL per mouse) were administered into the flank region of 7-week-old male C57BL/6 mice. Mice were observed for 3 to 4 weeks. Tumour growth was measured regularly using a digital calliper, and tumour volume (mm^3^) was estimated using the standard ellipsoid formula: $Volume\left(V\right)=Length\left(L\right)\times {Width(W)}^{2}\times 0.5$. Tumour weight (g) was recorded at the end of the experiments when the mice were culled, and the tumours were isolated.

### 4.3. Immunohistochemistry

Formalin-fixed, paraffin-embedded tumour tissues were sectioned into 5 μm slices using a LEICA RM2245 microtome (Leica Biosystems, Nussloch, Germany). Antigen retrieval was performed by boiling the tissue sections in 10 mM Tris-EDTA buffer (pH 9.0) for 30 min at 99 °C, followed by a 30 min cooling period at room temperature (Appendix A). Endogenous peroxidase activity was blocked by incubating the slides in Dako REAL™ peroxidase blocking solution (S2023, Agilent Technologies, Glostrup, Denmark) for 15 min in the dark. Non-specific binding was blocked using 5% normal goat serum (NGS) and 1% bovine serum albumin (BSA) in TBS-T (Appendix A) for 1 h at room temperature. The slides were then incubated overnight at 4 °C with primary antibodies listed in Appendix A. The following day, slides were incubated with goat anti-rabbit HRP-conjugated polymer (K4003, EnVision+ System-HRP (DAB) kit, K4003, Dako, Agilent Technologies, Glostrup, Denmark) for 1 h. The signals were developed using the EnVision FLEX DAB+ Substrate Chromogen System (K3468, Dako, Agilent Technologies, Glostrup, Denmark), followed by counterstaining with haematoxylin (S3309, Dako, Agilent Technologies, Glostrup, Denmark). Slides were subsequently rehydrated, mounted using DPX mounting medium (06522, Sigma-Aldrich, St. Louis, MO, USA), and air-dried for 24 h before analysis. Whole-slide imaging was carried out using the Aperio AT2 brightfield scanner (Leica Biosystems, Nussloch, Germany). Quantitation assessment was performed with HALO image analysis software v4.1 (Indica Labs, Albuquerque, NM, USA). For immune-related markers (CD3, CD4, CD8, CD103, CD11c, CD11b, CD86, CD40, and MHC-II) and the hypoxia marker HIF-1α, the number of positively stained cells was quantified and divided by the total number of cells. For vascular and adhesion markers (CD31, CD34, ICAM-1, VCAM-1), and MHC-I, the positively stained area was calculated and rationed to the total tissue area. A fixed intensity threshold was applied throughout all samples to maintain consistent staining criteria across all.

For measuring MVD, tumour sections were stained with haematoxylin and eosin (H&E). Red blood cell (RBC) clusters, indicative of vessel presence, were identified manually by an independent observer blinded to the sample groups. The percentage of MVD (MVD %) was calculated by dividing the total number of RBC clusters by the total stained tissue area.

### 4.4. Immunofluorescence

After antigen retrieval and blocking, the slides were incubated overnight at 4 °C with anti-CD31 antibody. After washing, sections were incubated with goat anti-rabbit HRP-conjugated secondary antibody for 1 h, followed by Alexa Fluor 488-conjugated tyramide for 10 min. After washing, a second round of antigen retrieval was performed. After blocking, the slides were incubated overnight at 4 °C with anti- NG2, or α-SMA antibodies (Appendix A). Slides were then incubated with HRP-conjugated secondary antibody for 1 h, followed by Alexa Fluor 594-conjugated tyramide for 10 min. All slides were counterstained with DAPI and mounted using VECTASHIELD antifade medium (H-1700, Vector Laboratories, Burlingame, CA, USA). Images were acquired using the Zeiss Axioscan 7 slide scanner (Carl Zeiss AG, Oberkochen, Germany) and analysed using HALO software v4.1 (Indica Labs, Albuquerque, NM, USA) with the fluorescent area quantification module. Thresholds were adjusted to standardise quantification across each sample.

### 4.5. Multiplex Immunohistochemistry (mIHC)

After antigen retrieval and blocking, mIHC was performed using a sequential tyramide signal amplification (TSA)-based protocol. Primary antibodies were applied one at a time, each diluted in 10% NGS and 1% BSA in TBS-T. The slides were incubated with each antibody at 4 °C in a humidified dark chamber overnight. After each primary antibody incubation, slides were washed in TBS-T, followed by incubation with an HRP-conjugated secondary antibody for 1 h at room temperature in the dark. The fluorescent signal was developed using the corresponding tyramide-conjugated Alexa Fluor dye for 10 min.

Following signal development, antigen retrieval was performed to remove bound antibodies while preserving the covalently deposited fluorophores. This cycle (blocking, primary and secondary antibody incubation, fluorophore development, and antigen retrieval) was repeated sequentially for each marker. The staining order and associated tyramide-conjugated dyes were: CD103 (AF594), CD8 (AF647), CD31 (AF555), and ICAM-1 (AF488) (Appendix A). After completing the final cycle of the staining rounds, sections were counterstained with Spectral DAPI (FP1490, Akoya Biosciences, Marlborough, MA, USA) diluted 1:500 in PBS for 5 min to visualise nuclei. The slides were then mounted with antifade mounting medium.

Multiplex-stained slides were imaged using the PhenoImager HT system (Akoya Biosciences, Marlborough, MA, USA) at 20× magnification. Exposure times were optimised by changing tyramide-conjugated fluorophores to maintain a dynamic range between 50 and 250 milliseconds. Spectral unmixing of fluorescent signals was performed using InForm software v3.1 (Akoya Biosciences, Marlborough, MA, USA) to separate overlapping emission spectra. The unmixed images were used for quantitative analysis of marker expression and spatial distribution in the tumour tissues.

### 4.6. Western Blot

WT, PAK1KO, and PAK4KO cells were seeded in 24-well plates and cultured for 48 h. Cells were then lysed in 2× loading buffer (Appendix A). Protein samples were separated using 10% SDS-PAGE gels and transferred to nitrocellulose membranes. Membranes were probed with primary antibodies against MHC-I, MHC-II, ICAM-1, VCAM-1, PAK1, PAK4, and GAPDH (Appendix A). After incubation with HRP-conjugated goat anti-rabbit IgG (Bio-Rad), signals were developed using ECL Select™ Detection Reagent (RPN2235, Cytiva, Amersham, UK) and imaged with a ChemiDoc™ MP Imaging System (Bio-Rad Laboratories, Hercules, CA, USA). Band density for each blot was quantified using ImageJ (Java 1.8.0_322).

### 4.7. Proteomics Analysis

Proteomic analysis was performed as previously described [46], including sample preparation, liquid chromatography-mass spectrometry using data-independent acquisition (DIA), and database searching. Briefly, KPC WT, PAK1KO, and PAK4KO tumour tissues were flash-frozen and ground into fine powder using an ice-cooled mortar and pestle. Tissue lysates were prepared in RIPA buffer containing protease and phosphatase inhibitors (Appendix A), followed by acetone precipitation and enzymatic digestion into peptides. Liquid chromatograph data independent acquisition mass spectrometry, and database search were carried out according to the established protocol [53].

For bioinformatic analysis, raw mass spectrometry data were processed using the Perseus software platform (v2.1.3.0). For global proteomic analysis, label-free quantification (LFQ) intensity values were log_2_-transformed and filtered to retain proteins with at least three valid values per group. Differential protein expression was assessed using a two-sample *t*-test, with statistical significance defined by an S_0_ value of 0.1 and a false discovery rate (FDR) < 0.05.

A volcano plot was generated using the same parameters, plotting-log Student’s *t*-test *p*-values on the *y*-axis and the log_2_fold-change (student *t*-test difference between group) on the *x*-axis. Significantly altered proteins were highlighted. Protein–protein interaction (PPI) networks were generated using the STRING app (v2.2.0) in Cytoscape (version 3.10.3), based on Mus musculus database searches with a confidence score cutoff of 0.7. Functional enrichment and pathway mapping were performed across all categories, and pathways of interest were selected. A continuous colour scale was used to visualise significantly upregulated proteins in red colour and downregulated proteins in blue colour.

### 4.8. Statistical Analysis

Quantitative data were presented as mean ± standard error of the mean (SEM). For comparisons between two groups, unpaired two-tailed Student’s *t*-tests were used, assuming Gaussian distribution and equal variances. For analyses involving two independent variables, two-way ANOVA was conducted with the inclusion of an interaction term, followed by Sidak’s multiple comparisons test, with a single pooled variance. A 95% confidence interval was used throughout, and *p*-values < 0.05 were considered statistically significant. All statistical analyses were performed using GraphPad Prism version 10.3.0 (GraphPad Software, San Diego, CA, USA).

## Figures and Tables

**Figure 1 ijms-26-08357-f001:**
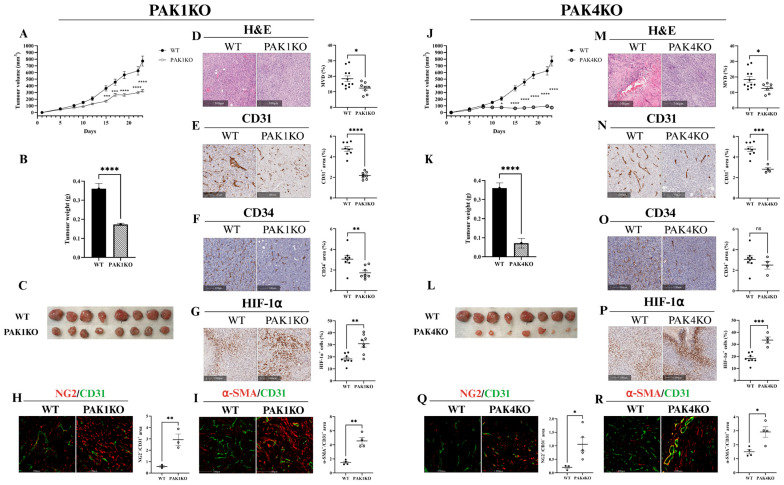
PAK1 or PAK4 knockout suppressed pancreatic tumour growth by reducing angiogenesis and increasing vascular normalisation. KPC WT (*n* = 11), PAK1KO (*n* = 8) or PAK4KO (*n* = 8) were subcutaneously injected into the flanks of C57BL/6 mice and tumour growth was monitored as described in the Materials and Methods. PAK1KO or PAK4KO significantly suppressed tumour growth by decreasing tumour volume and weight (**A**,**B**,**J**,**K**), and visibly smaller tumours (**C**,**L**). The H&E staining of tumour tissues showed reduced microvessel density (MVD) in PAK1KO (**D**) and PAK4KO (**M**) tumours. PAK1KO further reduced CD31 and CD34 expression (**E**,**F**). PAK4KO decreased CD31 expression (**N**) but not CD34 (**O**). The reduced angiogenesis by PAK1KO or PAK4KO suppressed tumour growth but also promoted hypoxia by increasing HIF-1α expression (**G**,**P**). However, knockout of PAK1 or PAK4 increased pericyte coverage marked by increased ratios of pericyte markers of NG2 or α-SMA to CD31 (**H**,**I**,**Q**,**R**), indicating enhanced vascular normalisation. Knockout of PAK1 or PAK4 reduced tumour angiogenesis while enhancing vascular normalisation to suppress tumour growth. WT: wild-type, KO: knockout, * *p* < 0.05, ** *p* < 0.01, *** *p* < 0.001, **** *p* < 0.0001, ns: not significant.

**Figure 2 ijms-26-08357-f002:**
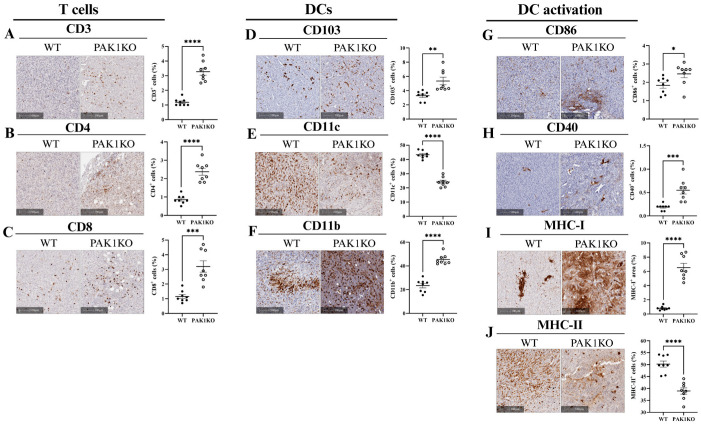
PAK1 knockout increased dendritic cell activation and promoted T-cell-mediated immune responses in pancreatic tumours. PAK1 knockout increased tumour-infiltrating T-cells, indicated by elevated CD3^+^ (**A**), CD4^+^ (**B**), and CD8^+^ T-cell populations (**C**). CD103^+^ dendritic cells, known to facilitate cross-priming of CD8^+^ T-cells, were also significantly increased in PAK1KO tumours (**D**), suggesting enhanced antigen presentation. While CD11c^+^ dendritic cells were reduced (**E**), a notable increase in CD11b^+^ dendritic cells was observed (**F**). PAK1KO further increased the expression of co-stimulatory molecules CD86 and CD40 (**G**,**H**), and MHC-I (**I**), reflecting dendritic cell maturation and activation. However, MHC-II expression was decreased (**J**), suggesting a selective modulation of antigen presentation. Collectively, these data indicate that PAK1 knockout reshaped the tumour immune microenvironment by enhancing dendritic cell function and promoting T-cell infiltration. WT: wild-type, KO: knockout, * *p* < 0.05, ** *p* < 0.01, *** *p* < 0.001, **** *p* < 0.0001.

**Figure 3 ijms-26-08357-f003:**
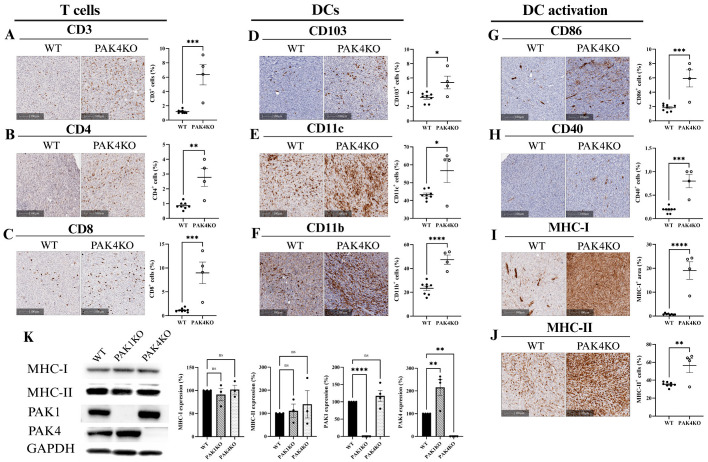
PAK4 knockout increased dendritic cell activation and promoted T-cell-mediated immunity in pancreatic tumours. PAK4 knockout increased the tumour-infiltration of CD3^+^ (**A**), CD4^+^ (**B**), and CD8^+^ T-cells (**C**). PAK4KO tumours also showed a marked increase in CD103^+^ dendritic cells (**D**), which are crucial for priming CD8^+^ T-cell responses. In contrast to PAK1KO tumours, both CD11c^+^ and CD11b^+^ dendritic cell subsets were significantly increased in the PAK4KO tumours (**E**,**F**), indicating a broader dendritic cell activation profile. Further, the expression of CD86 (**G**), CD40 (**H**), MHC-I (**I**), and MHC-II (**J**) was also strongly upregulated, indicating enhanced dendritic cell maturation and activation. The levels of MHC-I and MHC-II in WT, PAK1KO, or PAK4KO cancer cells were the same (**K**). These findings collectively highlighted a key immunoregulatory role for PAK4 in suppressing dendritic cell activation and T-cell function in the pancreatic tumour microenvironment. Protein quantification was performed using ImageJ, and all comparisons were made relative to wild-type controls, set as 100%. WT: wild-type, KO: knockout, * *p* < 0.05, ** *p* < 0.01, *** *p* < 0.001, **** *p* < 0.0001, ns: not significant.

**Figure 4 ijms-26-08357-f004:**
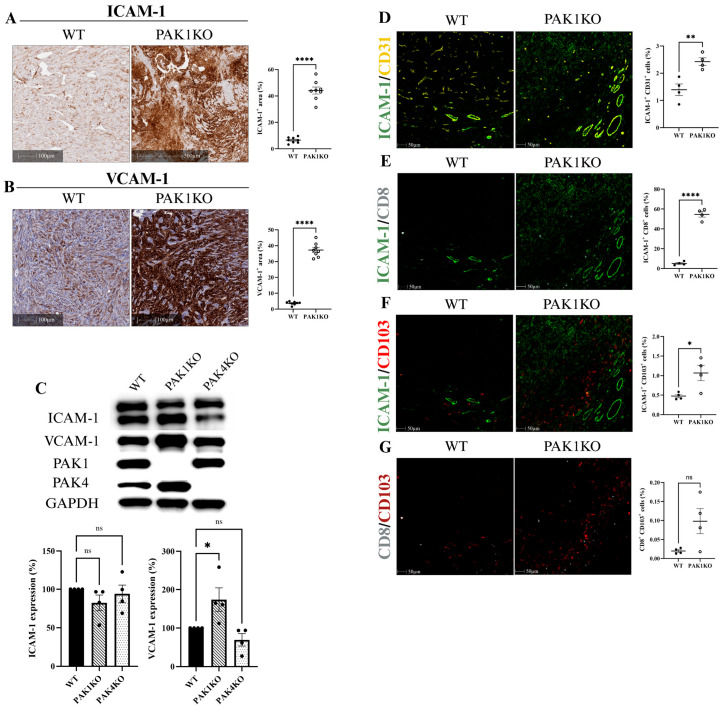
PAK1 knockout enhances T-cell infiltration/activation by upregulating endothelial adhesion molecules ICAM-1 and VCAM-1. The expression of ICAM-1 (**A**) and VCAM-1 (**B**) were increased by 7 times and 10 times in PAK1KO tumour tissues, respectively, indicating enhanced endothelial activation. The level of VCAM-1 in PAK1KO cancer cells was also elevated by less than 2 times (**C**) while the level of ICAM-1 remained unchanged across all cancer cell lines examined (**C**), suggesting that the upregulation of ICAM-1 and VCAM-1 in PAK1KO tumours was primarily within the stromal cells. Multiplex immunohistochemistry further demonstrated that PAK1KO increased ICAM-1 expression on CD31^+^ cells (**D**), CD8^+^ T-cells (**E**), and CD103^+^ dendritic cells (**F**), supporting a role for ICAM-1 in mediating immune cell adhesion and infiltration. Although not statistically significant, a trend toward increased CD8^+^CD103^+^ cells was observed (**G**). These findings implicated that PAK1 knockout activated the tumour vasculature to favour immune cell recruitment through adhesion molecule upregulation. Protein quantification was performed using ImageJ, and all comparisons were made relative to wild-type controls, set as 100%. WT: wild-type, KO: knockout, * *p* < 0.05, ** *p* < 0.01, **** *p* < 0.0001, ns: not significant.

**Figure 5 ijms-26-08357-f005:**
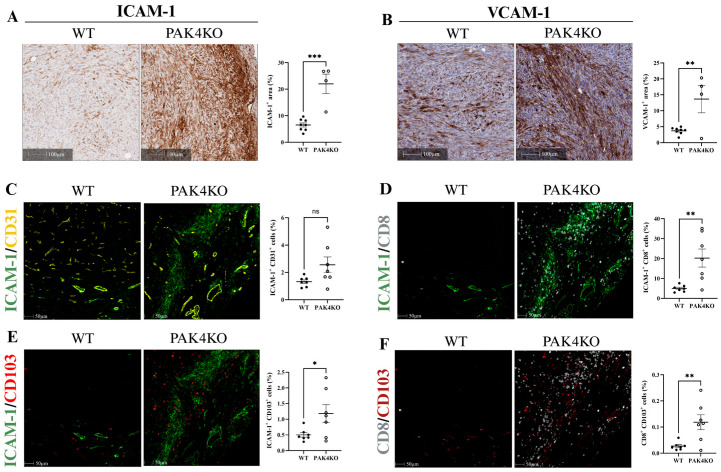
PAK4 knockout facilitated T-cell infiltration/activation by increasing ICAM-1 and VCAM-1 expression in the tumour microenvironment. The expression of ICAM-1 (**A**) and VCAM-1 (**B**) were increased in PAK4KO tumour tissues, but not in PAK4KO cancer cells (Figure 4C), suggesting enhanced potential for immune cell trafficking. Although ICAM-1 expression on CD31^+^ cells showed a non-significant upward trend (**C**), multiplex immunohistochemistry revealed a significant increase in ICAM-1^+^CD8^+^ T-cells (**D**) and ICAM-1^+^CD103^+^ dendritic cells (**E**), indicating active immune engagement with ICAM-1. Furthermore, the abundance of CD8^+^CD103^+^ cells, representing tissue-resident cytotoxic lymphocytes, was significantly higher in PAK4KO tumours (**F**), supporting enhanced immune cell infiltration and activation. These findings indicate that PAK4KO facilitated immune-vascular interactions and promoted immune cells’ accessibility by upregulating key adhesion molecules of ICAM-1. WT: wild-type, KO: knockout, * *p* < 0.05, ** *p* < 0.01, *** *p* < 0.001, ns: not significant.

**Figure 6 ijms-26-08357-f006:**
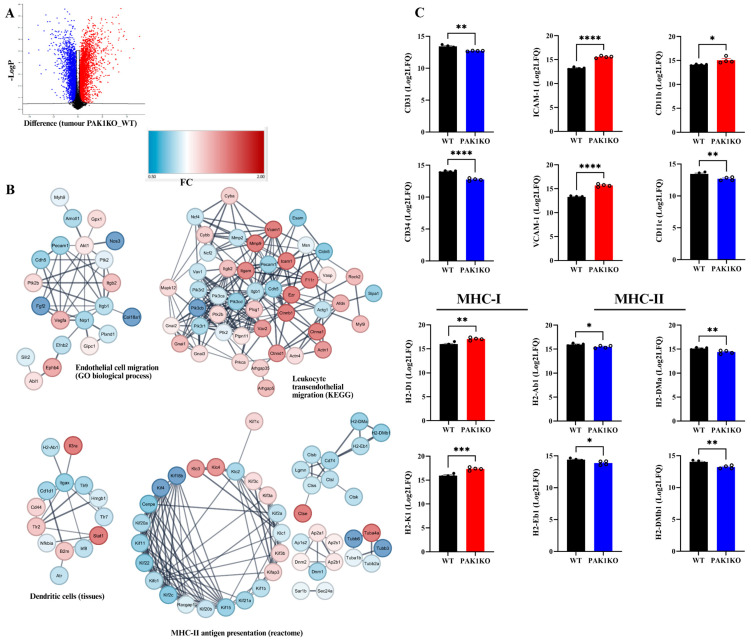
PAK1 knockout upregulated ICAM-1 and VCAM-1 and induced changes in leukocyte trans-endothelial migration. The tumour tissues from wild-type (WT) and PAK1-knockout (PAK1KO) were subjected to proteomic analysis. A volcano plot showed the differential expression of protein profiles in PAK1KO tumours compared to WT tumours (**A**). The red and blue colours represent up- and down-regulation of protein expression, respectively. Protein interaction network analysis of global proteomic data suggested significant alterations in processes involved in endothelial cell migration, dendritic cell function, and MHC-II antigen presentation, and enrichment of the KEGG leukocyte trans-endothelial migration pathway (**B**). The expression of ICAM-1 and VCAM-1, CD11b, and MHC class I molecules (H2-D1 and H2-K1) were increased while CD31 and CD34, CD11c, and MHC class II molecules (H2-Ab1, H2-Eb1, H2-DMa, and H2-DMb1) were significantly downregulated in PAK1KO tumours (**C**). *, *p* < 0.05, **, *p* < 0.01, ***, *p* < 0.001, ****, *p* < 0.0001.

**Figure 7 ijms-26-08357-f007:**
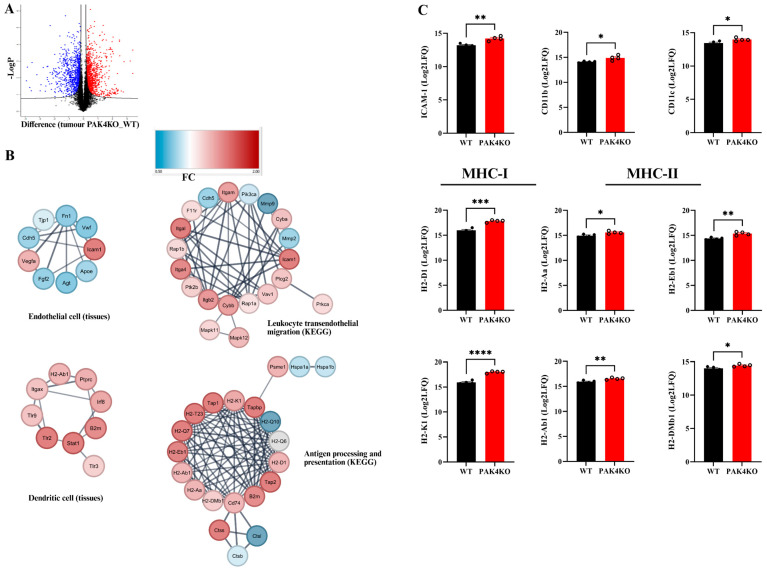
PAK4 knockout upregulated ICAM-1, CD11b, CD11c, and antigen presentations and induced changes in leukocyte trans-endothelial migration. The tumour tissues from wild-type (WT) and PAK4-knockout (PAK4KO) were subjected to proteomic analysis. A volcano plot showed the differential expression of protein profiles in PAK1KO tumours compared to WT tumours (**A**). The red and blue colours represent up- and down-regulation of protein expression, respectively. Protein interaction network analysis of global proteomic data suggested alterations in endothelial cell regulation, upregulation of dendritic cell activation pathway, and enrichment in KEGG pathways involved in leukocyte trans-endothelial migration and antigen processing and presentation (**B**). The expression of ICAM-1, CD11b, and CD11c were significantly higher in PAK4KO tumour (**C**). In addition, MHC class I molecules (H2-D1 and H2-K1) and MHC class II components (H2-Aa, H2-Ab1, H2-Eb1, and H2-DMb1) were significantly upregulated in PAK4KO tumour (**C**). *, *p* < 0.05, **, *p* < 0.01, ***, *p* < 0.001, ****, *p* < 0.0001.

## Data Availability

Data will be made available upon request from the corresponding author.

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
