# Peer review of "P-21 Kinase 1 or 4 Knockout Stimulated Anti-Tumour Immunity Against Pancreatic Cancer by Enhancing Vascular Normalisation"

_ijms, 2025, doi:10.3390/ijms26178357_

Round 1

Reviewer 1 Report

Comments and Suggestions for Authors

The manuscript presents interesting and solid data on the role of PAK1 and PAK4 depletion in modulating tumor vasculature, pericyte coverage, and immune cell infiltration. The experimental design is coherent, and the multi-parametric analyses provide a valuable contribution to understanding how PAK signaling may influence the tumor microenvironment.

That said, there are two conceptual aspects that I believe would significantly strengthen the work:

  1. VE-cadherin expression and phosphorylation status
    PAK kinases have been reported to regulate endothelial barrier integrity through the phosphorylation of VE-cadherin, particularly at Ser665 (PMID: 17060906), a modification linked to junction disassembly and leukocyte transmigration. Given the central role that endothelial junctions play in vascular “normalization” and immune cell extravasation, it would be valuable to assess VE-cadherin levels and, if possible, its phosphorylation at Ser665 in the experimental groups. This could be approached via immunoblotting of tumor-derived endothelial fractions or immunohistochemistry co-localized with CD31. Even if such data cannot be experimentally added at this stage, a discussion of this potential mechanism would help connect the observed phenotypes with known molecular pathways downstream of PAK.

  2. Broadening the angiogenesis perspective: vasculogenic mimicry and VE-cadherin modifications
    While the study primarily considers classical endothelial-driven angiogenesis, alternative vascularization mechanisms—particularly vasculogenic mimicry (VM)—are relevant to the context of pancreatic cancer, the tumor type investigated here. VM has been correlated with poor prognosis in pancreatic cancer and is closely associated with post-translational modifications of VE-cadherin, which have been extensively studied as drivers of VM formation. Given that the current manuscript already focuses on PAK, a kinase family with potential regulatory impact on VE-cadherin function, acknowledging VM and other non-classical angiogenesis processes (e.g., intussusceptive angiogenesis, vessel co-option) would broaden the scope and translational relevance of the work. Discussing whether PAK modulation might affect these mechanisms—particularly the VE-cadherin modifications linked to VM—along with including key references on VE-cadherin and VM in the pancreatic cancer literature, would integrate the findings into a more comprehensive angiogenesis framework.

  3. Overall, I support the use of the term “vascular normalization” in the manuscript. However, considering the multifaceted nature of tumor angiogenesis, an expanded discussion that goes beyond the traditional sprouting model, and which is anchored in the relevant VE-cadherin/VM literature, would enhance the depth and impact of the study.

Author Response

Reviewer 1

Comments and Suggestions for Authors

The manuscript presents interesting and solid data on the role of PAK1 and PAK4 depletion in modulating tumor vasculature, pericyte coverage, and immune cell infiltration. The experimental design is coherent, and the multi-parametric analyses provide a valuable contribution to understanding how PAK signaling may influence the tumor microenvironment.

That said, there are two conceptual aspects that I believe would significantly strengthen the work:

1.VE-cadherin expression and phosphorylation status

PAK kinases have been reported to regulate endothelial barrier integrity through the phosphorylation of VE-cadherin, particularly at Ser665 (PMID: 17060906), a modification linked to junction disassembly and leukocyte transmigration. Given the central role that endothelial junctions play in vascular “normalization” and immune cell extravasation, it would be valuable to assess VE-cadherin levels and, if possible, its phosphorylation at Ser665 in the experimental groups. This could be approached via immunoblotting of tumor-derived endothelial fractions or immunohistochemistry co-localized with CD31. Even if such data cannot be experimentally added at this stage, a discussion of this potential mechanism would help connect the observed phenotypes with known molecular pathways downstream of PAK.

Thank you for this constructive comment and valuable suggestions. We agree with the reviewer that VE-cadherin, a key component of endothelial adhesion junction, regulates endothelial barrier function and leukocyte transmigration. Thus, it is valuable to assess VE-cadherin and its phosphorylation status in the experiment groups. This will be a key area in our future study to investigate how PAKs affect endothelial junctions in cancer settings. Accordingly, we have added a paragraph on page 10 (the 3rd paragraph in the clean copy of the revised manuscript) to discuss the involvement of PAK in regulation of VE-cadherin and VM. We have also added a Fig. S3 in the Supplementary Materials.

  1. Broadening the angiogenesis perspective: vasculogenic mimicry and VE-cadherin modifications

While the study primarily considers classical endothelial-driven angiogenesis, alternative vascularization mechanisms—particularly vasculogenic mimicry (VM)—are relevant to the context of pancreatic cancer, the tumor type investigated here. VM has been correlated with poor prognosis in pancreatic cancer and is closely associated with post-translational modifications of VE-cadherin, which have been extensively studied as drivers of VM formation. Given that the current manuscript already focuses on PAK, a kinase family with potential regulatory impact on VE-cadherin function, acknowledging VM and other non-classical angiogenesis processes (e.g., intussusceptive angiogenesis, vessel co-option) would broaden the scope and translational relevance of the work. Discussing whether PAK modulation might affect these mechanisms—particularly the VE-cadherin modifications linked to VM—along with including key references on VE-cadherin and VM in the pancreatic cancer literature, would integrate the findings into a more comprehensive angiogenesis framework.

Thank you for the valuable and constructive suggestion. We greatly appreciate the suggestion to add discussion on VM and VE-cadherin to broaden the angiogenesis perspective. We have added one paragraph on page 10 (the 3rd paragraph in the clean copy of the revised manuscript) to address this point. We have added additional data from proteomic study on the changes of protein expressions related to VM formation including VE-cadherin (CHD5), CD34, EphA2, MMP14, and MMP2, and proposed a mechanism showing how downregulation of CDH5, EPHA2, and CD34 suppresses the MMP2/MMP14 axis, decreasing cancer-cell derived VM and contributing to vessel normalization.

  1. Overall, I support the use of the term “vascular normalization” in the manuscript. However, considering the multifaceted nature of tumor angiogenesis, an expanded discussion that goes beyond the traditional sprouting model, and which is anchored in the relevant VE-cadherin/VM literature, would enhance the depth and impact of the study.

Thank you for these constructive comments. Please refer to the answers to Q1 and Q2 above.

In summary, we have made the changes below to the manuscript.

  1. Replace Fig 1H, 1I, 1Q and 1R as the “(%)” units were removed from each Y-axis of the quantification data.
  2. Add the quantitation data of Western blots to Fig.3K and Fig.4C as requested by the reviewer.
  3. In the Supplementary Material, replaced Supplementary Fig.1 with tumour weight normalised by the mouse body weight as requested by the reviewer.
  4. Add supplementary figures 3,4, and 5 to the supplementary material to address the reviewers’ concerns.
  5. Add “Clone name” to each antibody through supplementary Tables 2 to 5 as requested by the reviewer.

Reviewer 2 Report

Comments and Suggestions for Authors

Ansardamavandi et al., in their manuscript entitled “P-21 kinase 1 or 4 knockout stimulates anti-tumour immunity against pancreatic cancer by enhancing vascular normalization”, present a well-written study. The figures are comprehensive and supported by detailed tissue analyses, and I found the manuscript engaging to review. I do, however, have a few suggestions that could further strengthen the work:

  • All western blot data quantification can be moved from suppl to the mani figures to their respective data to improve the figure panels for more readability.
  • The lymphoid cell–specific findings are particularly interesting. However, while CD11b⁺ immature monocytes appear increased in both PAK1 and PAK4 KO tumor tissues, it would be important to determine and discuss whether Ly6C⁺ monocytes differ between these models, as this could help define distinct myeloid versus lymphoid cell contributions.
  • For Figures 1C and 1L, please include tumor weight data normalized to mouse body weight.
  • I think its relevant and would be valuable to discuss whether PAK1 plays a role in endothelial cell migration/adhesion? Since its role as pathological migration is highly relevant in this disease context, particularly in endothelial cells. The authors may wish to reference prior literature (e.g., PMID: 25388666) in the discussion.
  • The ICAM1 and VCAM1 results are intriguing; could the authors also assess whether soluble forms of these adhesion molecules differ in serum? Similarly, evaluation of serum VEGF levels would be informative.
  • Many immunostaining panels appear to lack isotype controls, which makes it difficult to fully interpret the staining specificity.
  • Please provide full details for clone numbers of the antibodies used in this study.

Author Response

Reviewer 2

Comments and Suggestions for Authors

Ansardamavandi et al., in their manuscript entitled “P-21 kinase 1 or 4 knockout stimulates anti-tumour immunity against pancreatic cancer by enhancing vascular normalization”, present a well-written study. The figures are comprehensive and supported by detailed tissue analyses, and I found the manuscript engaging to review. I do, however, have a few suggestions that could further strengthen the work:

  1. All western blot data quantification can be moved from suppl to the mani figures to their respective data to improve the figure panels for more readability.

Thank you for this constructive suggestion. We have now moved the Western blot quantification graphs from the Supplementary materials into the main figures: Fig.3K and Fig.4 panels alongside the corresponding blots to improves figure readability and allow a clearer interpretation of the quantitative differences across experimental groups. Fig.S1 has now been replaced with tumour weight data normalized with mouse body weight to answer question 3.

  1. The lymphoid cell–specific findings are particularly interesting. However, while CD11b⁺ immature monocytes appear increased in both PAK1 and PAK4 KO tumor tissues, it would be important to determine and discuss whether Ly6C⁺ monocytes differ between these models, as this could help define distinct myeloid versus lymphoid cell contributions.

Our proteomic analysis revealed an increase in Ly6c1 expression in PAK4KO tumours, whereas PAK1 deletion did not change the expression of Ly6C (Fig. S4). These findings suggest distinct myeloid dynamics across the models: PAK4 deletion preferentially enriched for Ly6C⁺ monocytes. Notably, Ly6C⁺ monocytes are functionally associated with CD11c⁺CD11b⁺Ly6C⁺ populations, which act as key antigen-presenting cells supporting antitumour immune responses. This is now added to the end of the first paragraph on page 11 (the clean copy of the revised manuscript), Discussion section.

  1. For Figures 1C and 1L, please include tumor weight data normalized to mouse body weight.

The data of tumour weight normalized to mouse body weight has now been presented in Fig.S1.

  1. I think its relevant and would be valuable to discuss whether PAK1 plays a role in endothelial cell migration/adhesion? Since its role as pathological migration is highly relevant in this disease context, particularly in endothelial cells. The authors may wish to reference prior literature (e.g., PMID: 25388666) in the discussion.

We thank the reviewer for this valuable suggestion. PAK1 has been shown to regulate endothelial migration and adhesion, processes that underpin pathological angiogenesis in cancer. Prior studies, highlight PAK1’s role in cytoskeletal remodelling and focal adhesion turnover, thereby promoting endothelial motility and sprouting angiogenesis. In our study, PAK1 was depleted in tumour cells rather than endothelial cells. The reduced vessel density observed in PAK1KO tumours likely reflects impaired paracrine signalling from cancer cells that normally activate PAK1-dependent endothelial pathways. A few sentences are added to the end of second paragraph on page 10 (Discussion section).

  1. The ICAM1 and VCAM1 results are intriguing; could the authors also assess whether soluble forms of these adhesion molecules differ in serum? Similarly, evaluation of serum VEGF levels would be informative.

We appreciate the reviewer’s suggestion to evaluate soluble forms of ICAM1, VCAM1, and VEGF in serum, which could provide additional insight into systemic changes associated with PAK1 or PAK4 knockout. However, there were not enough serum sample collected to assess soluble forms of ICAM-1, VCAM-1 and VEGF for this study. We shall include these assessments in our future study to determine whether these tissue-level alterations are reflected in circulating soluble factors.

  1. Many immunostaining panels appear to lack isotype controls, which makes it difficult to fully interpret the staining specificity.

We acknowledge the relevance of isotype controls in validating immunostaining specificity. In this study, we employed rabbit IgG isotype controls for two representative antibodies, CD31 and HIF-1α, as all primary antibodies used were raised in rabbit (Fig. S5). These isotype controls consistently showed no detectable signal, thereby confirming the specificity of our staining. Given the uniform antibody host (rabbit) across all panels and the reproducibility of these negative control results, we did not repeat isotype controls for each additional marker.

  1. Please provide full details for clone numbers of the antibodies used in this study.

The antibody tables (Tables S2-S5) have now been updated to include clone names for all antibodies where the information was available from the manufacturers.

In summary, we have made the changes below to the manuscript.

  1. Replace Fig 1H, 1I, 1Q and 1R as the “(%)” units were removed from each Y-axis of the quantification data.
  2. Add the quantitation data of Western blots to Fig.3K and Fig.4C as requested by the reviewer.
  3. In the Supplementary Material, replaced Supplementary Fig.1 with tumour weight normalised by the mouse body weight as requested by the reviewer.
  4. Add supplementary figures 3,4, and 5 to the supplementary material to address the reviewers’ concerns.
  5. Add “Clone name” to each antibody through supplementary Tables 2 to 5 as requested by the reviewer.

Round 2

Reviewer 1 Report

Comments and Suggestions for Authors

The authors have responded to the two points I asked about. For my part, I would like to thank the authors for their dedication to them.

Reviewer 2 Report

Comments and Suggestions for Authors

Authors have thoroughly revised the manuscript based on my comments. Thanks.